# What information is preserved in latent cell embeddings?
# A Benchmark for Single-Cell Reconstruction

**Xiaotong Fu**[1,2]  **Dominik Klein**[1,2]  **Alessandro Palma**[1,2]
**Egor Antipov**[1,2]  **Alejandro Tejada Lapuerta**[1,2]  **Fabian J. Theis**[1,2*]
[1]Helmholtz Munich    [2]Technical University of Munich

## Abstract

Learning compressed representations of single-cell transcriptomics data has been instrumental in modeling biological and experimental shifts in cellular states. Most current methods embed cells into a low-dimensional representation, traverse the latent space in meaningful biological directions, and decode back to gene expression. Despite its importance, the choice of representation is typically treated as an implementation detail rather than a first-order modeling decision. However, if a latent space fails to capture biological information, the resulting cell embeddings may be biologically implausible, thus limiting modeling efficacy and downstream analysis. We therefore study a necessary requirement of representations for latent shift modeling: reconstruction, i.e., decoding latent representations back to gene expression profiles. Here, we present a systematic benchmark of reconstruction quality across widely used representation families (PCA, AEs, VAEs) and pretrained foundation-model embeddings augmented with trained decoders (scGPT, SCimilarity, STATE, scConcept). Across three datasets spanning perturbational and observational settings and different scales, we quantify reconstruction performance using both statistical fidelity and biological signal preservation, providing an empirical foundation for selecting representation schemes that retain connections to interpretable expression-based biological information.

## 1 Introduction

Thanks to its transcriptome-wide readout at an unprecedented resolution, the single-cell RNA-seq (scRNA-seq) technology has revolutionized our understanding of key aspects in cellular biology, such as state heterogeneity, disease programs, and cellular dynamics (Haque et al., 2017). However, scRNA-seq measurements are sparse, noisy, and high-dimensional, making many downstream tasks challenging in the gene space. Thus, it is crucial to learn a lower-dimensional cell representation that retains biological variation while compressing noise, enabling perturbation modeling, cell type annotation, and rare population discovery (Lähnemann et al., 2020; Lotfollahi et al., 2022).

Recently, a central emerging goal in single-cell modeling is the virtual cell: predictive models that simulate how a cell's transcriptome changes under internal or external interventions, including drug treatment, cytokine stimulation, or disease progression (Bunne et al., 2024). Many state-of-the-art perturbation methods achieve this goal by modeling shifts in a latent cell space and then decoding the shifted representations back to gene expression. For example, CellFlow uses generative modeling to predict perturbations in compressed data representations (Klein et al., 2025), while STATE matches predicted and ground-truth populations by minimizing their energy distance in a latent space (Adduri et al., 2025). This latent-shift paradigm is especially appealing because it reduces dimensionality, enables flexible conditional generation, often simplifying perturbation modeling and improving computational efficiency (Lotfollahi et al., 2019; Bunne et al., 2022; 2023; Tong et al., 2023; Eyring et al., 2023; Uscidda & Cuturi, 2023; Klein et al., 2024).

Most latent-shift studies default to standard embeddings (e.g., PCA or scVI), typically treating the representation as given rather than explicitly validating its suitability for gene-level generation

---

*Correspondence to `fabian.theis@helmholtz-munich.de`

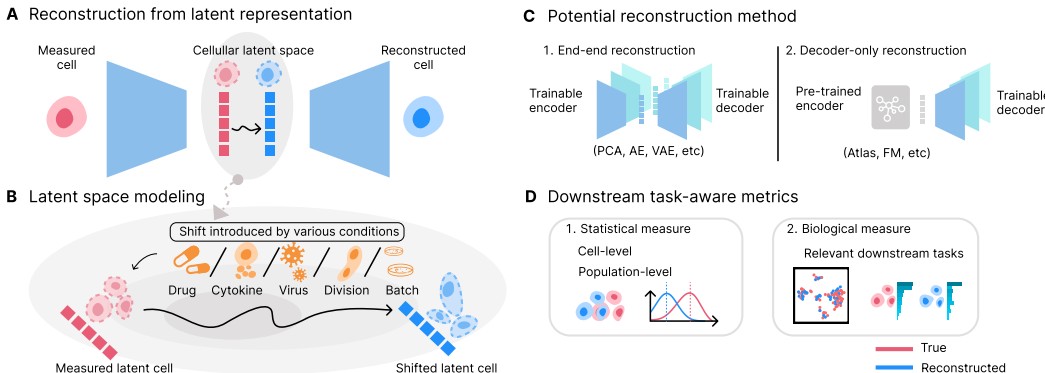

Figure 1: **Concept overview of single cell reconstruction benchmarking. A**) Reconstruction maps a latent representation back to an expression profile. **B**) A key use case is latent space shift modeling, where conditions induce shifts in the latent space that must decode back to gene-level changes. **C**) We benchmark two reconstruction schemes: I) End-to-end models with a jointly trained encoder and decoder, and II) decoder-only models that reconstruct from a fixed pretrained embedding using a trainable decoder. **D**) We evaluate reconstruction using both statistical fidelity and downstream task-aware biological metrics to assess whether the latent representation retains biological information.

(Ahlmann-Eltze & Huber, 2025; Klein et al., 2025). However, the representation can be a fundamental bottleneck: if a latent space does not retain sufficient information to reconstruct realistic gene expression, then decoded cells after a latent shift may be implausible even when the latent transformation is well fit. This is of utmost importance in the virtual cell context because biological interpretation (i.e., differential expression, pathway analysis, and gene-gene structure) is often defined in the gene expression space rather than abstract latent space.

This question becomes even more pressing with the rise of single-cell foundation models (Theodoris et al., 2023; Cui et al., 2024; Hao et al., 2024). Pretrained encoders provide general-purpose embeddings that perform well across a range of discriminative and retrieval tasks, yet many are decoder-free and are rarely evaluated for gene-level reconstruction. As a result, it remains unclear how much gene-resolution information these embeddings retain, and whether they can serve as latent spaces for perturbation modeling when coupled with a learned decoder.

Motivated by these gaps, we benchmark widely used single-cell representations with respect to reconstructive performance. We evaluate classical methods of encoder–decoder models (PCA, AEs, VAEs), and we additionally assess pretrained foundation-model embeddings by training decoders on top of fixed representations, enabling a unified comparison across representation families. We frame the core question as *reconstruction*: mapping a latent cell representation back into gene expression space to recover interpretable, gene-resolution profiles, and we quantify reconstruction quality using complementary statistical and biological metrics.

## 1.1 RELATED WORK

Many computational methods have been developed to address a variety of tasks in single-cell transcriptomics, and their relative performance can vary substantially across settings (Ziegenhain et al., 2017; Tian et al., 2019). As a result, benchmarking has become an essential tool for guiding method selection and establishing evaluation standards. For example, scIB serves as a comprehensive comparison of atlas-scale data integration methods (Luecken et al., 2022), while Wei et al. (2025) and Wenteler et al. (2024) benchmark deep learning models for perturbation prediction. Other efforts focus on single-cell trajectory inference approaches (Saelens et al., 2019; Inecik et al., 2025), cell type annotation (Crowley & Quake, 2025), and feature selection (Zappia et al., 2025). In this work, we focus on an under-evaluated component shared by many pipelines: the cell representation and its ability to support faithful reconstruction back to gene space.

Standard representation methods for scRNA-seq modeling typically rely on compressed representations learned via linear or non-linear encoder–decoder architectures (Eraslan et al., 2019). Among

Table 1: Benchmark datasets span distinct perturbation regimes (drug, cytokine, none), biological systems (cancer, immune, lung tumor), and scales ($\sim$1–100M cells), providing complementary diversity for evaluating representation learning for reconstruction.

| Dataset | Perturbation | Tissue/System | Setting | Cells | #Conditions |
|---|---|---|---|---|---|
| *Tahoe-100M* | Drugs (1137) | Pan-cancer | Cell lines (50) | 89,423,257 | $\sim$60k |
| *PBMC-10M* | Cytokines (90) | Blood (PBMC) | Primary (ex vivo) | 9,697,974 | $\sim$17k |
| *LuCA* | None (observational) | Lung | Patient atlas | 892,296 | $\sim$1k |

these, variational autoencoders (VAEs) are a dominant framework, with widely used models such as scVI (Lopez et al., 2018) and its extensions (Ashuach et al., 2023; Martens et al., 2024) offering principled links between latent representations and complex molecular readouts. More recently, the emergence of large-scale foundation models for cellular representation learning has introduced an alternative paradigm for deriving latent embeddings, including approaches based on metric learning (Heimberg et al., 2025) and transformer architectures (Bahrami et al., 2025; Cui et al., 2024; Adduri et al., 2025; Palla et al., 2025). In this work, we consider all of these representation learning strategies to assess which are most suitable when reconstruction accuracy is the primary objective.

## 1.2 CONTRIBUTION

In this work, we:

- Formalize reconstruction as a necessary property of latent spaces for latent shift modeling.
- Propose a unified evaluation suite combining both statistical similarity and downstream-relevant biological signal preservation;
- Benchmark both classic encoder-decoder representation families and pretrained foundation-model embeddings on large-scale datasets spanning perturbational and observational regimes;
- Derive practical recommendations and identify trade-offs between latent spaces, enabling more principled representation choices for latent shift modeling.

Figure 1 summarizes our setup: (i) reconstruction as the key property of representations in latent shift modeling, (ii) two reconstruction paradigms, and (iii) a joint statistical–biological evaluation.

## 2 DATASETS AND TASKS

Cellular state shifts arise from both external perturbations (e.g., drug or cytokine exposure) and endogenous changes driven by microenvironmental or disease processes. To give a comprehensive overview of reconstructive performance across different regimes, we benchmark on three scRNA-seq datasets spanning perturbational and observational settings: *Tahoe-100M* (Zhang et al., 2025), *Parse PBMC-10M* (Parse Biosciences, 2026; Oesinghaus et al., 2025), and the *Single-Cell Lung Cancer Atlas (LuCA)* (Salcher et al., 2022). Together, they cover distinct perturbation modalities, biological systems, and scales, while providing substantial condition diversity (see Table 1). Detailed dataset descriptions and pre-processing are provided in Appendix A.1.

Here, we define a *condition* as the minimal unit of modeling and evaluation, instantiated as a combination of covariates such as (cell type/line × treatment × dose). To probe extrapolation, we construct multiple train/test splits per dataset with increasing out-of-distribution (OOD) difficulty (e.g., holding out entire cell types/lines, donors, or subsets of conditions). Split construction and statistics are described in Appendix A.1.

We evaluate reconstruction performance through proxies of common downstream analyses. Specifically, we assess: (I) recovery of perturbation effects via differential expression, (II) consistency of underlying biological programs via pathway activity, and (III) preservation of broader biological structure via gene co-expression patterns and cell-cycle composition. Metric definitions are summarized in Section 4.2, with full formulas in Appendix A.3.

## 3 METHODS

Let $X \in \mathbb{N}^M$ denote a multivariate random variable representing a cell-level gene expression vector over $M$ genes. Our goal is to evaluate representation learning methods through their ability to support accurate reconstruction of $X$ after compression into a low-dimensional latent $Z \in \mathbb{R}^d$, with $d < M$. We formalize each method with an encoder $g_\phi : \mathbb{N}^M \to \mathcal{P}(\mathbb{R}^d)$ that defines a posterior $q_\phi(Z \mid X)$ over latent variables, and a decoder $f_\theta : \mathbb{R}^d \to \mathcal{P}(\mathbb{N}^M)$ that defines a likelihood $p_\theta(X \mid Z)$. Reconstruction quality is assessed by comparing samples from $p_\theta(X \mid Z)$ (with $Z \sim q_\phi(Z \mid X)$) against the observed data, using metrics described in Section 4.2.

We distinguish two reconstruction schemes that reflect common practice in single-cell analysis:

- **End-to-end reconstruction:** both encoder and decoder are trained jointly on the training set, yielding a task-specific embedding $Z$.
- **Decoder-only reconstruction:** the encoder is fixed to, for example, a pretrained foundation model (FM). A separate decoder is trained on the training set to reconstruct $X$ from the fixed embedding.

We intend to use such separation to mirror how embeddings are actually obtained: classical latent models are trained on the dataset of interest, whereas FMs are typically used as frozen generalized representations with a small task-specific head.

### 3.1 END-TO-END RECONSTRUCTION

For end-to-end models, we consider representative baselines from both general machine learning and single-cell genomics: Principal Component Analysis (PCA), Autoencoder (AE), and Variational Autoencoder (VAE). Exact objectives and parameterizations are provided in Appendix A.2.

**PCA.** serves as a linear baseline, which yields the optimal rank-$d$ reconstruction in the least-squares sense (see Appendix A.2). We treat PCA as a reference point for both reconstruction fidelity and the benefits of nonlinear modeling.

**AE.** parameterizes $g_\phi$ and $f_\theta$ as deterministic deep neural networks trained to minimize a reconstruction objective. AEs have been widely used for representation learning in domains such as computer vision and speech (Huang et al., 2022; Zheng et al., 2025).

**VAE.** further defines a stochastic latent variable model with a Normal distribution prior. In single-cell genomics, VAE-based models (e.g., scVI) are widely adopted due to their probabilistic nature and interpolation ability (Xu et al., 2021; Lotfollahi et al., 2022).

**Library size handling.** Sequencing depth (library size) is a major technical factor that can be treated either as a technical covariate or as part of the biological signal, and different modeling choices can substantially affect both reconstruction and downstream use (Hafemeister & Satija, 2019). To isolate this axis, we evaluate three common strategies:

1. **None:** no library size information is provided to the model; the decoder must implicitly account for depth variation during reconstruction.

2. **Modeled:** library size is a learned cell-level variable inferred from $X$ during training.

3. **Observed:** true library size is provided explicitly as an input covariate to the decoder.

These settings probe whether a method's reconstruction performance is driven by modeling meaningful gene-gene structure versus exploiting explicit sequencing depth information. We detail the corresponding parameterizations in Appendix A.2.2.

### 3.2 DECODER-ONLY RECONSTRUCTION

For decoder-only models, we evaluate several widely used and recent single-cell foundation models as fixed encoders, including scGPT (Cui et al., 2024), SCimilarity (Heimberg et al., 2025), STATE (Adduri et al., 2025), and scConcept (Bahrami et al., 2025). For each FM, we follow the recommended preprocessing and embedding extraction protocol (details in Appendix A.2). This setting

reflects the dominant usage pattern of FMs in practice: generalized, pretrained embeddings transferred to a new task, here corresponding to gene expression signal reconstruction.

Given an embedding $Z$ from a frozen FM encoder, we train a decoder $f_\theta$ to reconstruct gene expression: A multilayer perceptron matching the architecture of end-to-end decoders (i.e., AE, VAE) (see Section 3.1). Decoders are trained from scratch on each dataset split using the same modeling and optimization settings as end-to-end models (Appendix A.4). This ensures that performance differences primarily reflect how much information about $X$ is retained in the frozen embedding.

## 4 BENCHMARKING

To enable a fair and informative comparison, we evaluate all methods within an aligned experimental design and report metrics that capture both statistical fidelity and biological signal preservation relevant to common downstream analyses. Concretely, we control key confounders (latent dimensionality, library-size handling, and training budget), evaluate generalization under multiple OOD regimes, and quantify reconstruction quality at both cell and population levels.

### 4.1 EXPERIMENT SETTING

**Latent dimensionality.** Since the latent dimension $d$ is a primary bottleneck for reconstruction and directly determines the capacity available for latent-space shift modeling, we benchmark a wide range of values: $d \in \{10, 32, 128, 512, 2048\}$, covering common settings in single-cell analysis as well as more extreme ones. Unless explicitly stated, comparisons across model families are made at matched $d$ to eliminate dimensionality as a confounder. Scaling behavior trends with $d$ are further analyzed in Appendix A.5.

**Library size handling.** For models that admit explicit sequencing-depth modeling (i.e., AE/VAE decoders), we evaluate the three library-size strategies defined in Section 3 (None / Modeled / Observed), while for the PCA-based decoder, we only consider direct expression count reconstruction.

**Hyperparameter optimization.** To avoid unfair advantages from ad hoc tuning, we define a preregistered hyperparameter search space (e.g., depth, width, and KL weight for VAEs) and conduct a grid search per experiment. For each method family and setting, we report the best-performing configuration on the validation set under this budget. Search spaces, budgets, and selection criteria are detailed in Appendix A.4.

**OOD evaluation.** For each dataset, we construct multiple train/test splits with different OOD difficulty (e.g., holding out cell types/lines vs. holding out subsets of conditions) as described in Appendix A.1.3. Each method is trained on the training set, selected on the validation set, and evaluated on the OOD test set without access to held-out conditions.

To summarize, we define an experiment as a fixed configuration of the type (dataset $\times$ split $\times$ latent dimension $d$); for each model family and library-size strategy, we run a predefined hyperparameter search and report the validation-selected best run under a fixed budget.

### 4.2 METRICS

Prior work often focuses on pure statistical measures such as MSE or MMD or limited summaries such as differential expression (DEG). Here, we construct a set of metrics that quantify statistical fidelity and biological signal preservation relevant to downstream analyses. Full definitions and implementation details are provided in Appendix A.3.

#### 4.2.1 STATISTICAL MEASURE

Single-cell measurements often exhibit substantial stochasticity and technical noise. Thus, per-cell correspondence between an observed cell and its reconstruction is generally ill-posed. We complement cell-level with population-level evaluation of the reconstruction output within each condition. More in detail, we report both:

**Cell-level reconstruction**. When a deterministic point reconstruction $\hat{X}$ is available, we compute the mean squared error (MSE) between $X$ and $\hat{X}$.

**Population-level reconstruction.** Let $\mathcal{X}$ and $\hat{\mathcal{X}}$ denote the sets of ground-truth and reconstructed cells for a condition (see Section 2). We compute: (I) mean-level agreement via $R^2$ between condition-wise mean expression vectors, and (II) higher-order distributional agreement using Maximum Mean Discrepancy (MMD) between $\mathcal{X}$ and $\hat{\mathcal{X}}$.

To reduce sensitivity to the high-dimensionality of gene expression space, we compute MMD in a shared low-dimensional evaluation space: we fit PCA on the full ground-truth data and project both $\mathcal{X}$ and $\hat{\mathcal{X}}$ onto the top 50 PCs before computing distances.

### 4.2.2 BIOLOGICAL MEASURE

We quantify whether reconstructions still preserve biologically meaningful variation through proxies of common downstream analyses:

**Perturbational effects.** We compare DE results between control and perturbed populations extracted from reconstructed and ground-truth data, reporting recall of top-ranked DE genes and correlation of log-fold-changes (logFC) across genes for each contrast.

**Biological program consistency.** We compute pathway activity scores through pre-defined responsive genes (Schubert et al., 2018) on reconstructed and ground-truth cells and report similarity (e.g., Pearson / Spearman correlation).

**Biological structure preservation.** We evaluate the preservation of gene-gene correlation structure by computing coexpression matrices for curated gene sets (i.e., MSigDB collections from Subramanian et al. (2005)) on reconstructed and ground-truth cells and report similarity (e.g., Pearson / Spearman correlation). In addition, we assess whether cell-cycle composition is preserved by computing canonical cell-cycle scores from established marker genes (Tirosh et al. (2016)) and comparing the distributions across phases between reconstructed and ground-truth cells.

### 4.2.3 AGGREGATED SCORE

To provide a robust summary of overall performance across heterogeneous metrics, we define a rank-based aggregate score. For each experiment of the type (dataset $\times$ split $\times$ $d$), we rank methods per metric, average ranks within statistical and biological categories, and then average these two category ranks to obtain a final aggregate rank. This aggregation reduces sensitivity to metric scaling and enables comparison of models that perform well across multiple aspects of reconstruction.

## 5 EXPERIMENTS

### 5.1 END-TO-END

We first summarize *end-to-end* reconstruction at latent dimension $d = 128$, which we use as a representative point on our latent space dimensionality grid ($d \in \{10, 32, 128, 512, 2048\}$). This choice avoids focusing on either severely bottlenecked or near-saturated regimes and provides a stable comparison point across model families. We report results on *Tahoe-100M*, *PBMC-10M*, and *LuCA*. Sensitivity to latent dimensionality and performance under different OOD splits are provided in Appendix A.5. Table 2 aggregates results over 3 datasets and 3 split settings ($n = 9$ experiments total). For each method and library-size strategy, we report the mean $\pm$ standard deviation across these experiments for each metric, together with the rank-based overall score defined in Section 4.2.3.

**Overall comparison.** Across metrics and datasets, AEs achieve the strongest overall performance, consistently ranking above scVI and PCA. AEs perform the best for most statistical and biological metrics, except for being slightly worse on DEG recall and LogFC correlation. PCA achieves the highest DEG recall, and scVI achieves the best logFC similarity. Varying library-size handling yields relatively modest changes compared to differences between model families. For AEs, the modeled/observed library size slightly improves aggregate performance, whereas scVI shows a performance drop under modeled library size. We therefore consider library-size strategy as a design choice specific to model architecture and dataset.

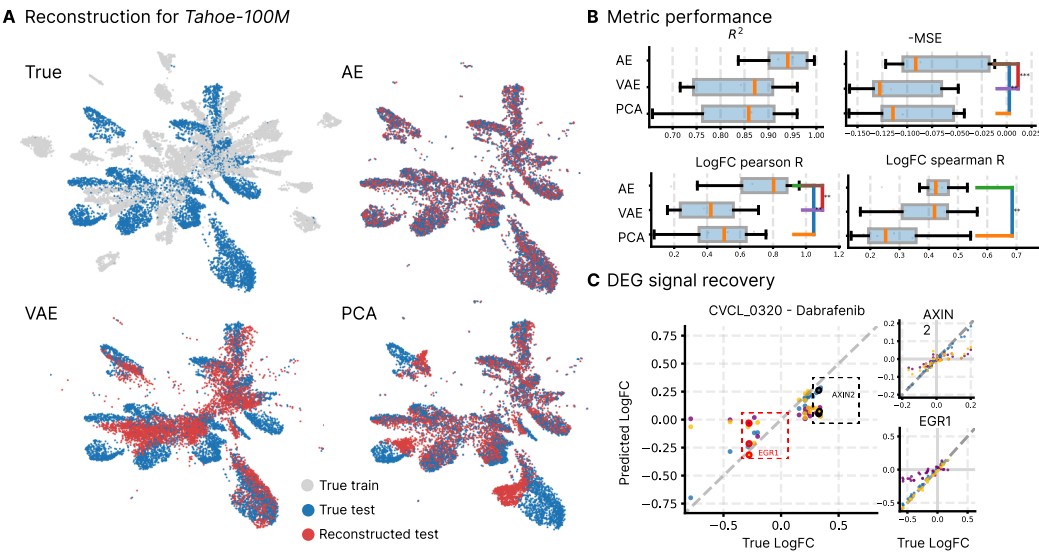

Figure 2: **Perturbation signal recovery on *Tahoe-100M* (cell-line OOD split). A**) Visualization of true training cells (gray), true held-out test cell lines (blue), and reconstructions of test cells (red) for AE, VAE, and PCA. **B**) Distribution of statistical and DEG-related metrics across all experiments; significance is assessed using paired $t$-test across matched experiments. We report -MSE so that a higher value is consistently better across metrics. **C**) Example drug (Dabrafenib) and maker genes (AXIN, EGR1) illustrating recovery of differential-expression effects both in drug and gene levels.

Table 2: End-to-end reconstruction performance summary (mean $\pm$ 1 std across dataset×split settings; $n = 9$ for Overall). Library size handling: (N) none, (M) modeled, (O) observed. We report MSE (lower is better); all other columns are higher-is-better. Bold indicates the best per column.

| | | Statistical | | | Biological | | | | |
|---|---|---|---|---|---|---|---|---|---|
| Model | Overall | MSE | $R^2$ | E-distance | Pathway Sp. | Coexpr. Sp. | Cell-cycle | DEG recall | LogFC Sp. |
| AE-N | 0.71 ±0.10 | 0.07 ±0.04 | 0.93 ±0.05 | 0.92 ±1 | 0.79 ±0.10 | **0.91 ±0.03** | 0.59 ±0.20 | 0.29 ±0.30 | 0.43 ±0.04 |
| AE-M | 0.78 ±0.05 | **0.07 ±0.04** | 0.93 ±0.06 | 0.93 ±1 | **0.79 ±0.10** | 0.90 ±0.03 | 0.61 ±0.20 | 0.28 ±0.30 | 0.43 ±0.04 |
| AE-O | **0.78 ±0.05** | 0.07 ±0.04 | **0.93 ±0.06** | **0.90 ±1** | 0.79 ±0.10 | 0.90 ±0.03 | **0.61 ±0.20** | 0.29 ±0.30 | 0.43 ±0.04 |
| scVI-N | 0.49 ±0.05 | 0.10 ±0.04 | 0.87 ±0.09 | 4.15 ±3 | 0.53 ±0.05 | 0.88 ±0.02 | 0.57 ±0.20 | 0.20 ±0.10 | 0.41 ±0.04 |
| scVI-M | 0.29 ±0.07 | 0.11 ±0.04 | 0.85 ±0.10 | 2.82 ±2 | 0.49 ±0.05 | 0.86 ±0.03 | 0.56 ±0.20 | 0.18 ±0.10 | 0.41 ±0.07 |
| scVI-O | 0.44 ±0.06 | 0.10 ±0.04 | 0.87 ±0.09 | 3.09 ±2 | 0.51 ±0.05 | 0.85 0.02 | 0.56 0.20 | 0.19 ±0.10 | **0.45 ±0.08** |
| PCA | 0.52 ±0.10 | 0.10 ±0.05 | 0.82 ±0.10 | 2.30 ±3 | 0.69 ±0.08 | 0.88 ±0.04 | 0.58 ±0.20 | **0.31 ±0.30** | 0.33 ±0.04 |

**Dataset-specific analysis.** We further present dataset-specific analyses. On *Tahoe-100M*, we focus on DEG-related metrics (DEG recall and logFC similarity; Figure 2) under the most challenging OOD split that holds out entire cell lines. Unless stated otherwise, we use the modeled library-size setting, which aligns more with real use cases where we try to infer library size from a shifted latent representation. AEs most faithfully reconstruct held-out cell lines while preserving perturbation signatures, recovering both logFC structure among DE genes and marker-gene shifts across drugs. Complementary analyses of pathway-level preservation on *PBMC-10M* and cell-cycle preservation on *LuCA* are provided in Appendix A.5: .

Overall, AEs provide the strongest reconstruction performance for end-to-end reconstruction, with consistent gains on both statistical and biological measures.

## 5.2 DECODER-ONLY

We next evaluate *decoder-only* reconstruction, where a pretrained foundation model encoder is kept frozen, and only a decoder is trained to map embeddings to gene expression (Section 3.2). To minimize leakage and better reflect transfer usage, we report results on *PBMC-10M*, which is not in

Table 3: Decoder-only reconstruction performance summary on PBMC-10M split by donor.

| Model | Decoder | Statistical | | | | Biological | | | |
|---|---|---|---|---|---|---|---|---|---|
| | | MSE | $R^2$ | E-distance | Pathway Sp. | Coexpr. Sp. | Cell-cycle | DEG recall | LogFC Sp. |
| STATE | MLP | **0.0923** | **0.9164** | 0.3967 | **0.7943** | **0.9551** | **0.4553** | 0.1207 | **0.4861** |
| scConcept | MLP | 0.1400 | 0.8645 | **0.2470** | 0.4860 | 0.8693 | 0.3567 | **0.1728** | 0.3524 |
| scGPT | MLP | 0.1408 | 0.8629 | 1.6563 | 0.4576 | 0.8824 | 0.3517 | 0.1653 | 0.3533 |
| SCimilarity | MLP | 0.1420 | 0.8607 | 0.7829 | 0.4544 | 0.8559 | 0.3449 | 0.1677 | 0.3361 |

any pre-training datasets of our model collections. Unless stated otherwise, we focus on the most challenging OOD split, holding out cell types. Results on additional splits are in Appendix A.5.

**Overall comparison.** Table 3 summarizes decoder-only reconstruction on *PBMC-10M*. Among the evaluated embeddings, STATE embedding with an MLP decoder achieves the strongest overall performance, except for E-distance and DEG recall. In contrast, scConcept shows complementary strengths, achieving the lowest E-distance (i.e., the closest reconstructed population distribution under this metric) and the highest DEG recall. An additional complication in decoder-only benchmarking is that pre-trained embeddings differ in dimensionality, which can confound reconstruction by changing the information bottleneck available to the decoder. We also compare decoder-only against end-to-end reconstruction to contextualize the embedding choice for latent-space shift modeling (Appendix A.5, Fig. 6).

**Dataset-specific analysis.** *PBMC-10M* captures cytokine-driven perturbations that manifest as coordinated transcriptional programs, we further examine pathway activity recovery. Figure 3 illustrates an example pathway (PI3K; scoring procedure in Appendix A.3.2) and compares reconstructed versus ground-truth activity patterns on held-out cell types. STATE achieves the best pathway recovery among decoder-only embeddings (highest pathway Spearman correlation), but still exhibits a noticeable gap relative to end-to-end reconstruction on *PBMC-10M* (Appendix A.5, Fig 4).

Overall, STATE provides the most promising embedding for reconstruction performance, with a gap still from the end-to-end reconstruction.

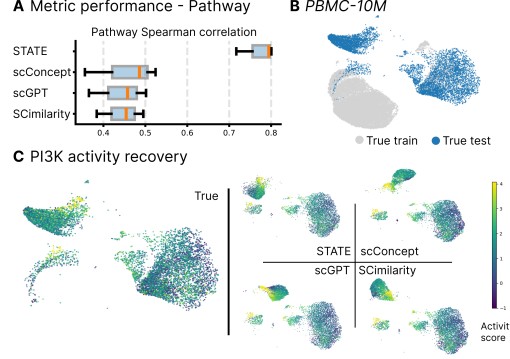

Figure 3: **Pathway recovery on *PBMC-10M* (cell-type OOD split). A**) Distribution of pathway recovery metric across all splits. **B**) Visualization of true training cells (gray), true held-out test cell lines (blue). **C**) Example pathway (PI3K) activity scores of different embeddings (right), showing pathway-level signal recovery.

## 6 DISCUSSION

**Summary.** In this work, we benchmarked the reconstruction quality of both widely-used single-cell representation models and foundation model embeddings. We compared (I) end-to-end encoder-decoder methods trained directly for reconstruction and (II) decoder-only settings where a frozen pretrained foundation-model embedding is paired with a trained decoder. Across datasets and OOD regimes, autoencoders consistently provided a promising reconstruction accuracy, while pretrained foundation-model embeddings were competitive on some signals but generally retained a gap to end-to-end training under our protocol.

**Limitations.** This benchmark focuses on reconstruction as a property of representations, rather than broader latent-space suitability for generative modeling (e.g., calibration, sampling quality, or intervention smoothness). Our results should therefore be read as reconstruction performance under controlled decoder capacity and training budgets—effectively an upper bound on what can be decoded from each representation family. While end-to-end embeddings outperform frozen foundation-model embeddings, an important next step is to evaluate fine-tuned foundation models and disentangle gains from pretraining scale versus dataset adaptation. Finally, we only consider log-normalized expression; extending to count-based decoders (e.g., negative binomial likelihoods) would clarify whether probabilistic count modeling yields benefits beyond log-space reconstruction.

**Future extension.** We envision this benchmark as a future task within the *Open Problems in Single-Cell Analysis* platform, providing a novel perspective on single-cell embedding methods and enabling more standardized, comparable evaluations across methods (Luecken et al., 2025).

MEANINGFULNESS STATEMENT

A meaningful representation of life should preserve the causal and compositional structure of biology: it should retain gene-level signals that define cell state and remain interpretable under interventions (e.g., perturbations or disease progression). In single-cell genomics, this implies embeddings that support faithful reconstruction and preserve downstream readouts such as differential expression and pathway programs. We contribute by formalizing reconstruction as a key property of latent spaces and benchmarking classical and foundation-model embeddings with metrics spanning statistical fidelity and biological utility. This makes representation choice more principled and advances latent shift modeling for intervention prediction.

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

# A APPENDIX

## A.1 DATA

### A.1.1 DATASET SUMMARY

**Dataset 1 (*Tahoe-100M*).** *Tahoe-100M* is a large-scale cancer cell line perturbation atlas measuring transcriptional responses to small-molecule treatments (Zhang et al., 2025). It contains 50 cancer cell lines and 379 drugs tested at three dosages, plus one vehicle control (DMSO), yielding 56,877 unique conditions (cell line $\times$ drug $\times$ dose) and 89,423,257 cells in total. Following the dataset creators, we exclude a technical replicate plate from all experiments to avoid potential leakage across splits; all reported statistics are computed after this exclusion.

**Dataset 2 (*PBMC-10M*).** *PBMC-10M* is a cytokine perturbation dataset profiling immune responses in peripheral blood mononuclear cells (Parse Biosciences, 2026; Oesinghaus et al., 2025). It contains 12 donors and 17 annotated cell types treated with 90 cytokines plus a PBS control, resulting in 17,030 unique conditions (cell type $\times$ donor $\times$ cytokine) and 9,697,974 cells.

**Dataset 3 (*LuCA*).** *LuCA* is a human lung cancer atlas integrating multiple NSCLC studies (Salcher et al., 2022). The dataset comprises 892,296 cells across 33 cell types and 1,030 unique conditions (cell type $\times$ study $\times$ tissue origin).

### A.1.2 PREPROCESSING

We apply a standardized preprocessing pipeline aligned with common practice in general latent shift modeling in single-cell analysis.

**End-to-end preprocessing.**

**Normalization.** For all datasets, we perform library-size normalization to a fixed target count followed by log transformation. Specifically, let $s_i = \sum_{g=1}^{M} x_{ig}$ be the library size (total counts) of cell $i$. We perform count-depth (library-size) normalization to a fixed target sum $T = 10^4$ followed by a log transform:

$$\tilde{x}_{ig} = \frac{T}{s_i} x_{ig}, \tag{1}$$

$$y_{ig} = \log(1 + \tilde{x}_{ig}). \tag{2}$$

This corresponds to `scanpy.pp.normalize_total(adata, target_sum=1e4)` and `scanpy.pp.log1p(adata)`.

**Highly variable gene selection.** To focus on biologically informative variation while keeping a reasonable computational costs, we subset each dataset to a curated highly variable gene set. For *Tahoe-100M*, we perform batch-aware HVG selection following established benchmarking practice (Luecken et al., 2022): we select the top 3,000 variable genes per plate, and retain genes present in at least two plates. We further include canonical cell-cycle marker genes (Tirosh et al., 2016) to ensure cell-cycle metrics are computable, yielding 6,087 genes in total. For *PBMC-10M*, we follow the dataset-recommended HVG preprocessing (Parse Biosciences, 2026), resulting in 5,412 genes. For *LuCA*, we use the predefined HVG set provided with the integrated atlas, resulting in 5,983 genes. All models are trained and evaluated on the corresponding dataset-specific gene set.

**Decoder-only preprocessing.**

**SCimilarity.** We follow the SCimilarity preprocessing utilities to (i) align gene ordering to the model's fixed feature space and (ii) apply the log-normalized TP10k procedure used during training (`align_dataset` and `lognorm_counts`). (Heimberg et al., 2025)

**STATE.** We follow the official STATE preprocessing interface, which contains only `log1p`. (Adduri et al., 2025)

**scGPT and scConcept.** For scGPT and scConcept, inputs are discretized via value binning and/or rank-based encodings as part of their modeling pipeline (Cui et al., 2024; Bahrami et al., 2025). Therefore, we follow the same preprocessing scheme defined in end-to-end reconstruction.

Table 4: Train/validation/test splits for each dataset at three OOD levels. Split 1 is the most OOD and Split 3 the least OOD. Counts are reported as train/val/test.

| Dataset | Split | OOD level | Unit held out | Condition definition | Counts (train/val/test) |
|---------|-------|-----------|---------------|----------------------|-------------------------|
| *Tahoe-100M* | Split 1 | High | Cell line | – | 34 / 8 / 8 cell lines |
| | Split 2 | Medium | Drug (incl. DMSO) | – | 266 / 57 / 57 drugs |
| | Split 3 | Low | Condition | Cell line $\times$ drug $\times$ dose | 39,813 / 8,532 / 8,532 conditions |
| *PBMC-10M* | Split 1 | High | Cell type | – | 11 / 3 / 3 cell types |
| | Split 2 | Medium | Donor | – | 8 / 2 / 2 donors |
| | Split 3 | Low | Condition | Cell type $\times$ donor $\times$ cytokine | 11,920 / 2,555 / 2,555 conditions |
| *LuCA* | Split 1 | High | Cell type | – | 23 / 5 / 5 cell types |
| | Split 2 | Medium | Study | – | 14 / 3 / 4 studies |
| | Split 3 | Low | Condition | Cell type $\times$ study $\times$ tissue origin | 720 / 155 / 155 conditions |

### A.1.3  DATA SPLITTING

We construct three train/validation/test splits per dataset to probe generalization at different levels of OOD. We refer to the splits from most to least OOD as **Split 1**, **Split 2**, and **Split 3**. To keep split definitions transparent and reproducible, all splits are defined using pre-existing dataset covariates (e.g., cell type, donor, drug). Unless stated otherwise, we use a train/validation/test ratio of 0.70/0.15/0.15 at the level of the held-out unit. See Table 4

### A.2  MODELS

### A.2.1  END-TO-END MODELS

We formulate end-to-end models in a unified latent-variable framework on log-transformed expression. Let $C \in \mathbb{N}^M$ denote raw UMI counts for a cell over $M$ genes, and let $X \in \mathbb{R}^M$ denote the log-normalized expression after the preprocessing in Section A.1.2. A model $\mathcal{M}$ consists of:

- an encoder $g_\phi : \mathbb{R}^M \to \mathcal{P}(\mathbb{R}^d)$ defining $q_\phi(Z \mid X)$,
- a decoder $f_\theta : \mathbb{R}^d \to \mathcal{P}(\mathbb{R}^M)$ defining $p_\theta(X \mid Z)$.

**Gaussian observation model.**  To align with common practice in latent shift modeling on continuous expression, we assume an isotropic Gaussian observation model on $X$:

$$p_\theta(X \mid Z) = \mathcal{N}\big(X; \mu_\theta(Z), \sigma^2 I_M\big), \tag{3}$$

where $\mu_\theta(\cdot)$ is the decoder mean (implemented by a neural network or linear map), and $\sigma^2$ is either fixed (i.e., AE) or learned (i.e., VAE).

**PCA.**  PCA can be viewed as a linear deterministic encoder–decoder with $Z \in \mathbb{R}^d$:

$$Z = W^\top(X - \bar{X}), \qquad W \in \mathbb{R}^{M \times d}, \quad W^\top W = I_d, \tag{4}$$

$$\hat{X} = \bar{X} + WZ = \bar{X} + WW^\top(X - \bar{X}), \tag{5}$$

where $\bar{X}$ is the empirical mean over training cells. Equivalently, for a data matrix $\mathbf{X}_{1:N} \in \mathbb{R}^{N \times M}$ of mean-centered rows, PCA solves the best rank-$d$ approximation problem:

$$\min_{\text{rank}(A) \le d} \|\mathbf{X}_{1:N} - A\|_F^2, \tag{6}$$

whose solution is given by truncating the SVD to the top $d$ components (Eckart–Young–Mirsky). (Eckart & Young, 1936)

**AE.**  For deterministic encoders as in AEs, the approximate posterior degenerates to a point mass:

$$q_\phi(Z \mid X) = \delta\left(Z - g_\phi(X)\right), \tag{7}$$

where $\delta(\cdot)$ denotes the Dirac delta. The reconstruction mean is

$$\hat{X} \triangleq \mathbb{E}_{Z \sim q_\phi(\cdot \mid X)}[\mu_\theta(Z)] = \mu_\theta\big(g_\phi(X)\big). \tag{8}$$

Training minimizes the negative log-likelihood under Eq. equation 3:

$$\mathcal{L}_{\text{AE}}(X; \theta, \phi) = -\log p_\theta(X \mid Z = g_\phi(X))$$

$$= \frac{1}{2\sigma^2} \|X - \mu_\theta(g_\phi(X))\|_2^2 + \text{const.} \tag{9}$$

Thus, with fixed $\sigma^2$, maximizing the Gaussian likelihood is equivalent (up to a constant factor) to minimizing MSE.

**VAE.** For probabilistic encoders as in VAEs, we use a diagonal Gaussian posterior:

$$q_\phi(Z \mid X) = \mathcal{N}\big(Z; \mu_\phi(X), \text{diag}(\sigma_\phi^2(X))\big), \tag{10}$$

$$p(Z) = \mathcal{N}(Z; 0, I_d). \tag{11}$$

The VAE is trained by minimizing the negative evidence lower bound (ELBO): (Kingma & Welling, 2013)

$$\mathcal{L}_{\text{VAE}}(X; \theta, \phi) = \mathbb{E}_{Z \sim q_\phi(Z|X)}[-\log p_\theta(X \mid Z)] + D_{\text{KL}}(q_\phi(Z \mid X) \parallel p(Z)). \tag{12}$$

The reconstructed expression $\hat{X}$ is again by equation 8. We also consider a weighted-KL variant ($\beta$-VAE) by replacing the KL term in equation 12 with $\beta D_{\text{KL}}(\cdot\|\cdot)$. (Higgins et al., 2017)

We adopt scVI from Lopez et al. (2018) as VAE implementation.

### A.2.2 LIBRARY SIZE HANDLING

Even after log-normalization, cells can exhibit substantial variation in sequencing depth ("library size"), which affects reconstruction difficulty and may confound downstream latent shift modeling. Throughout, models operate on log-normalized expression $X \in \mathbb{R}^M$ and first produce decoder outputs $\hat{X} \in \mathbb{R}^M$ (Eq. equation 8). We define a library-size proxy on the same scale as

$$L \triangleq \sum_{m=1}^{M} X_m. \tag{13}$$

Let $\tilde{X}$ denote the final model prediction used for evaluation. We consider three library-size modes:

1. **None.** No library size information is provided to the decoder. The model reconstructs directly in log-expression space, and we set

$$\tilde{X} = \hat{X}. \tag{14}$$

   In this mode, depth variation must be captured implicitly by the latent $Z$ and decoded back into $\hat{X}$.

2. **Modeled.** The library size is treated as a learned cell-level variable inferred from $X$ during training and then used to rescale the reconstructed profile:

$$\tilde{X} = \text{softmax}(\hat{X}) \cdot \hat{L}. \tag{15}$$

   For AEs, we predict $\hat{L}$ from $X$ using a lightweight one-layer head (a single linear layer). For VAEs (scVI), we introduce an additional latent variable for library size with a Gaussian prior $p(L)$ whose mean and variance are estimated from the training set, and use a variational posterior $q_\phi(L \mid X)$. The objective adds a KL term for $L$:

$$\mathcal{L}_{\text{VAE}}(X; \theta, \phi) + D_{\text{KL}}(q_\phi(L \mid X) \parallel p(L)), \tag{16}$$

   in addition to the standard ELBO terms in Eq. equation 12. For $\beta$-VAE variants, we apply the same KL weight $\beta$ to both KL terms.

3. **Observed.** The per-cell library size $L$ is computed from the input and provided explicitly to the decoder as a covariate. The decoder outputs $\hat{X}$, and the final prediction is obtained by allocating total mass according to $L$:

$$\tilde{X} = \text{softmax}(\hat{X}) \cdot L. \tag{17}$$

   This allows the model to represent depth variation via $L$ rather than encoding it into $Z$.

### A.3 METRICS

Let $\mathcal{X} = \{X_i\}_{i=1}^n$ and $\hat{\mathcal{X}} = \{\hat{X}_j\}_{j=1}^n$ denote the sets of ground-truth and reconstructed cells for a given condition, where each $X_i, \hat{X}_j \in \mathbb{R}^M$ is log-normalized expression (Section A.1.2).

#### A.3.1 STATISTICAL MEASURES

**MSE (cell-level).** When a paired reconstruction $\hat{X}_i$ is available for each observed cell $X_i$, we compute the mean squared error over cells and genes:

$$\text{MSE}(\mathcal{X}, \hat{\mathcal{X}}) = \frac{1}{nM} \sum_{i=1}^n \left\| X_i - \hat{X}_i \right\|_2^2. \tag{18}$$

$R^2$ **(population level).** Let $s(\mathcal{X}) = \sum_{i=1}^n X_i \in \mathbb{R}^M$ and $s(\hat{\mathcal{X}}) = \sum_{j=1}^{\hat{n}} \hat{X}_j \in \mathbb{R}^M$ denote condition-wise summed expression vectors . We report the coefficient of determination between these two $M$-dimensional vectors:

$$R^2(\mathcal{X}, \hat{\mathcal{X}}) = 1 - \frac{\left\| s(\mathcal{X}) - s(\hat{\mathcal{X}}) \right\|_2^2}{\left\| s(\mathcal{X}) - \overline{s(\mathcal{X})}\mathbf{1} \right\|_2^2}, \tag{19}$$

where $\overline{s(\mathcal{X})} = \frac{1}{M} \sum_{m=1}^M s(\mathcal{X})_m$ and $\mathbf{1} \in \mathbb{R}^M$ is the all-ones vector.

**MMD (population level).** For distributional distances (MMD), we first map cells into a shared low-dimensional evaluation space using PCA fit on all ground-truth cells:

$$\bar{X}_i = P(X_i) \in \mathbb{R}^{50}, \qquad \bar{\hat{X}}_j = P(\hat{X}_j) \in \mathbb{R}^{50}. \tag{20}$$

We compute Maximum Mean Discrepancy in the PCA space using an RBF kernel $k(u,v) = \exp\left(-\frac{\gamma}{2}\|u-v\|_2^2\right)$ with fixed $\gamma$:

$$\text{MMD}^2(\mathcal{X}, \hat{\mathcal{X}}) = \mathbb{E}_{i,i'}\left[k(\bar{X}_i, \bar{X}_{i'})\right] + \mathbb{E}_{j,j'}\left[k(\bar{\hat{X}}_j, \bar{\hat{X}}_{j'})\right] - 2\,\mathbb{E}_{i,j}\left[k(\bar{X}_i, \bar{\hat{X}}_j)\right], \tag{21}$$

where expectations are approximated by empirical averages over all pairs.

#### A.3.2 BIOLOGICAL MEASURES.

For each evaluated condition (treatment) we additionally require a matched reference (control) population. Let $\mathcal{R} = \{R_\ell\}_{\ell=1}^{n_{\text{ctrl}}}$ denote the ground-truth reference cells (e.g., DMSO / PBS / normal), and let $\hat{\mathcal{R}} = \{\hat{R}_\ell\}_{\ell=1}^{\hat{n}_{\text{ctrl}}}$ denote their reconstructed counterparts when available. We compute differential expression (DE) between treatment and reference for both ground-truth and reconstructed data.

**Differential expression operator.** Let

$$\text{DE}(\mathcal{A}, \mathcal{B}; \text{method}) := \left\{ \left( p_m(\mathcal{A}, \mathcal{B}),\, \Delta_m(\mathcal{A}, \mathcal{B}) \right) \right\}_{m=1}^M \tag{22}$$

return, for each gene $m$, an adjusted $p$-value $p_m$ and an estimated log-fold-change $\Delta_m$ (e.g., from a Wilcoxon rank-sum test as in `scanpy.tl.rank_genes_groups`). We define the top-$K$ DE gene set by adjusted $p$-value as

$$S_K(\mathcal{A}, \mathcal{B}) := \underset{m \in [M]}{\arg \text{topK}} \left( -p_m(\mathcal{A}, \mathcal{B}) \right) = \{\text{the } K \text{ genes with smallest } p_m(\mathcal{A}, \mathcal{B})\}. \tag{23}$$

**DEG recall@100.** Let $(p_m^\star, \Delta_m^\star) := \text{DE}(\mathcal{X}, \mathcal{R}; \text{method})$ denote ground-truth DE between observed treatment and observed reference, and $(\hat{p}_m, \hat{\Delta}_m) := \text{DE}(\hat{\mathcal{X}}, \hat{\mathcal{R}}; \text{method})$ denote DE between reconstructed treatment and reconstructed reference ("pred" setting). We report overlap-based recall at $K{=}100$:

$$\text{DEGRecall@100}(\mathcal{X}, \mathcal{R}, \hat{\mathcal{X}}, \hat{\mathcal{R}}) = \frac{1}{100} \left| S_{100}(\mathcal{X}, \mathcal{R}) \cap S_{100}(\hat{\mathcal{X}}, \hat{\mathcal{R}}) \right|. \tag{24}$$

(If both sets have size 100, this equals the Dice overlap used in our implementation up to a constant: Dice@100 = $2|A \cap B|/(|A| + |B|) = |A \cap B|/100$.)

**LogFC Spearman correlation.** Using the same DE outputs, we compute Spearman correlation between ground-truth and reconstructed log-fold-changes over the set of genes present in both results (typically all $M$ genes):

$$\rho_{\text{LogFC}} \;=\; \text{Spearman}\Big(\{\Delta_m^\star\}_{m \in \mathcal{G}},\; \{\hat{\Delta}_m\}_{m \in \mathcal{G}}\Big), \qquad \mathcal{G} = \{m \in [M] : \Delta_m^\star \text{ and } \hat{\Delta}_m \text{ are defined}\}.$$
(25)

Equivalently, Spearman correlation can be written as Pearson correlation on ranks:

$$\text{Spearman}(a, b) \;=\; \text{Pearson}(\text{rank}(a),\, \text{rank}(b)).$$
(26)

**Pseudo-code (per condition).**

```
Input: treatment cells X, reference cells R,
       reconstructed treatment X_hat, reconstructed reference R_hat,
       DE method, K = 100

# p-values and logFC for ground truth
(p*, lfc*)    = DE(X,     R;    method)

# "pred" setting
(p_hat, lfc^) = DE(X_hat, R_hat; method)
S*   = topK_genes_by_smallest_p(p*, K)
S^   = topK_genes_by_smallest_p(p_hat, K)

DEGRecall@K = |S* INTERSECT S^| / K

# correlate logFC across shared genes G
LogFC_Sp    = SpearmanCorr(lfc*[G], lfc^[G])
```

**Pathway activity Spearman correlation.** Let $\mathcal{P}$ be a fixed collection of pathways (we use PROGENy). Each pathway $p \in \mathcal{P}$ is associated with a weighted gene set (a regulatory "network") $\mathcal{N}_p = \{(g, w_{p,g})\}$ from the PROGENy model.

**Gene filtering and valid pathways.** Given ground-truth and reconstructed cells for the same condition, we first restrict evaluation to genes that are sufficiently expressed in the *ground-truth* cells (as in our implementation):

$$\mathcal{G} \;:=\; \{g \in [M] \;:\; \#\{i \in [n] : X_{i,g} > 0\} \;\geq\; n_{\min}\},$$
(27)

where $n_{\min}$ is the minimum number of cells in which a gene is detected. We then keep only pathways with enough overlap with $\mathcal{G}$:

$$\mathcal{P}_{\text{valid}} \;:=\; \big\{p \in \mathcal{P} \;:\; \big|\{g : (g, w_{p,g}) \in \mathcal{N}_p\} \cap \mathcal{G}\big| \;\geq\; \tau\big\},$$
(28)

where $\tau$ is an overlap threshold.

**ULM pathway scoring.** Let $\text{ULM}(\cdot; \mathcal{N})$ denote the univariate linear model scoring operator (as in `decoupler.mt.ulm`), which maps an expression matrix to per-cell pathway activity scores. Applying ULM to the (gene-filtered) true and reconstructed expression yields:

$$S \;=\; \text{ULM}(\{X_i\}_{i=1}^n;\, \{\mathcal{N}_p\}_{p \in \mathcal{P}_{\text{valid}}}) \in \mathbb{R}^{n \times |\mathcal{P}_{\text{valid}}|},$$
(29)

$$\hat{S} \;=\; \text{ULM}\Big(\{\hat{X}_i\}_{i=1}^n;\, \{\mathcal{N}_p\}_{p \in \mathcal{P}_{\text{valid}}}\Big) \in \mathbb{R}^{n \times |\mathcal{P}_{\text{valid}}|},$$
(30)

where columns correspond to pathways and rows correspond to cells.

**Per-pathway Spearman correlation across cells.** For each valid pathway $p \in \mathcal{P}_{\text{valid}}$, let $S_{:,p} \in \mathbb{R}^n$ and $\hat{S}_{:,p} \in \mathbb{R}^n$ be the vectors of pathway scores over cells. We compute:

$$\rho_p \;=\; \text{Spearman}\Big(S_{:,p},\, \hat{S}_{:,p}\Big), \qquad p \in \mathcal{P}_{\text{valid}}.$$
(31)

**Aggregation.** We report the average pathway correlation:

$$\text{PathwaySpearman}(\mathcal{X}, \hat{\mathcal{X}}) = \frac{1}{|\mathcal{P}_{\text{valid}}|} \sum_{p \in \mathcal{P}_{\text{valid}}} \rho_p. \qquad (32)$$

**Pseudo-code (per condition).**

```
Input: true cells X (n×M), recon cells X_hat (n×M),
       PROGENy network N_p for each pathway p in P,
       gene_min_cells = n_min, overlap_threshold = tau

# gene filter (as implemented: based on true data)
G = { g : count_i [X[i,g] > 0] >= n_min }

# select pathways with enough gene overlap
P_valid = { p in P : |genes(N_p) INTERSECT G| >= tau }
N_valid = { N_p : p in P_valid }   # subset PROGENy model

# compute per-cell pathway scores with ULM
S      = ULM(X[:,G],     N_valid)   # n × |P_valid|
S_hat  = ULM(X_hat[:,G], N_valid)   # n × |P_valid|

# per-pathway Spearman across cells
for p in P_valid:
    rho[p] = SpearmanCorr(S[:,p], S_hat[:,p])

PathwaySpearman = mean_p rho[p]
```

**Coexpression Spearman similarity.** Let $\mathcal{G}$ be a collection of gene sets (we use MSigDB Hallmark gene sets), where each gene set $G \in \mathcal{G}$ is a subset of genes. Given ground-truth and reconstructed cells for one condition, we quantify how well the reconstructed data preserves within-gene-set co-expression patterns.

**Gene filtering and valid gene sets.** We first define the set of genes expressed in at least $n_{\min}$ cells in the *ground-truth* data:

$$\mathcal{F} := \{ g \in [M] \, : \, \#\{ i \in [n] : X_{i,g} > 0 \} \geq n_{\min} \}. \qquad (33)$$

(As in our implementation, we take $\mathcal{F}$ from ground-truth only.) A gene set is considered valid if it overlaps sufficiently with $\mathcal{F}$:

$$\mathcal{G}_{\text{valid}} := \{ G \in \mathcal{G} \, : \, |G \cap \mathcal{F}| \geq \tau \}, \qquad (34)$$

where $\tau$ is an overlap threshold.

**Within-set Spearman correlation matrices.** Fix a valid gene set $G \in \mathcal{G}_{\text{valid}}$ and let $d_G := |G \cap \mathcal{F}|$. Let $X^{(G)} \in \mathbb{R}^{n \times d_G}$ and $\hat{X}^{(G)} \in \mathbb{R}^{n \times d_G}$ be the ground-truth and reconstructed expression restricted to genes in $G \cap \mathcal{F}$. We compute gene–gene Spearman correlation matrices across cells:

$$C_{ab}^{(G)} = \text{Spearman}\left( X_{:,a}^{(G)}, X_{:,b}^{(G)} \right), \qquad \hat{C}_{ab}^{(G)} = \text{Spearman}\left( \hat{X}_{:,a}^{(G)}, \hat{X}_{:,b}^{(G)} \right), \qquad a, b \in [d_G]. \qquad (35)$$

**Thresholding gene pairs (ground-truth only).** We optionally ignore weakly coexpressed pairs in the ground-truth by a magnitude threshold $\delta \geq 0$:

$$\mathcal{U}_G := \left\{ (a, b) : 1 \leq a < b \leq d_G, \ |C_{ab}^{(G)}| \geq \delta \right\}. \qquad (36)$$

**Pairwise similarity and gene-set score.** For each retained pair $(a, b) \in \mathcal{U}_G$, we convert correlation differences to a similarity in $[0, 1]$:

$$S_{ab}^{(G)} = 1 - \frac{\left| C_{ab}^{(G)} - \hat{C}_{ab}^{(G)} \right|}{2}, \qquad (a, b) \in \mathcal{U}_G, \qquad (37)$$

and define the coexpression score for gene set $G$ by averaging over retained pairs:

$$\text{Coexpr}(G) = \frac{1}{|\mathcal{U}_G|} \sum_{(a,b)\in\mathcal{U}_G} S_{ab}^{(G)}. \tag{38}$$

**Aggregation across gene sets.** Finally, we average across valid gene sets:

$$\text{CoexpressionSpearman}(\mathcal{X}, \hat{\mathcal{X}}) = \frac{1}{|\mathcal{G}_{\text{valid}}|} \sum_{G\in\mathcal{G}_{\text{valid}}} \text{Coexpr}(G). \tag{39}$$

**Pseudo-code (per condition).**

```
Input: true cells X (n×M), recon cells X_hat (n×M),
       gene sets {G} (e.g., Hallmark), n_min, overlap tau, pair threshold delta

# expressed genes in true data
F = { g : count_i [X[i,g] > 0] >= n_min }

# valid gene sets
G_valid = { G : |G INTERSECT F| >= tau }

scores = []
for G in G_valid:
    genes = list(G INTERSECT F)
    XG     = X[:, genes]
    XG_hat = X_hat[:, genes]

    # gene{gene Spearman correlation across cells
    C     = SpearmanCorrMatrix(XG)
    C_hat = SpearmanCorrMatrix(XG_hat)
    U = {(a,b): a<b and |C[a,b]| >= delta}
    if U is empty: continue / set NaN

    # similarity and gene-set score
    S_ab = 1 - |C[a,b]-C_hat[a,b]|/2  for (a,b) in U
    scores.append(mean_{(a,b) in U} S_ab)

CoexpressionSpearman = mean(scores)
```

**Cell-cycle phase agreement.** We evaluate whether reconstructed cells preserve cell-cycle phase assignments obtained from canonical S and G2M marker genes (Tirosh et al., 2016). Let $\mathcal{S}$ and $\mathcal{G}\in\mathcal{M}$ denote the S-phase and G2M-phase gene lists, respectively.

**Gene filtering.** As in our implementation, we restrict marker genes to those expressed in at least $n_{\min}$ cells in the ground-truth population:

$$\mathcal{F} := \{g \in [M] : \#\{i \in [n] : X_{i,g} > 0\} \geq n_{\min}\}, \qquad \mathcal{S}' = \mathcal{S} \cap \mathcal{F}, \quad \mathcal{G}\in\mathcal{M}' = \mathcal{G}\in\mathcal{M} \cap \mathcal{F}. \tag{40}$$

**Phase labeling operator.** Let $\Phi(\cdot; \mathcal{S}', \mathcal{G}\in\mathcal{M}')$ denote the phase labeling function induced by `scanpy.tl.score_genes_cell_cycle`, which maps each cell to a discrete phase:

$$y_i = \Phi(X_i; \mathcal{S}', \mathcal{G}\in\mathcal{M}') \in \{\text{G1}, \text{S}, \text{G2M}\}, \qquad \hat{y}_i = \Phi(\hat{X}_i; \mathcal{S}', \mathcal{G}\in\mathcal{M}') \in \{\text{G1}, \text{S}, \text{G2M}\}. \tag{41}$$

**Same-phase proportion (cell-level).** When reconstructions are paired cell-wise (i.e., $n = \hat{n}$), we compute the fraction of cells whose predicted phase matches the ground-truth phase:

$$\text{SamePhase}(\mathcal{X}, \hat{\mathcal{X}}) = \frac{1}{n} \sum_{i=1}^{n} \mathbf{1}[y_i = \hat{y}_i]. \tag{42}$$

**Pseudo-code (per condition).**

```
Input: true cells X (n×M), recon cells X_hat (n×M),
       S genes S, G2M genes G2M, n_min

# filter marker genes by detection in true data
F    = { g : count_i [X[i,g] > 0] >= n_min }
S'   = S   INTERSECT F
G2M' = G2M INTERSECT F
if |S'| < 5 or |G2M'| < 5: return NaN

# assign phases using Scanpy scoring
y     = PhaseLabels( X,     S', G2M')   # values in {G1,S,G2M}
y_hat = PhaseLabels( X_hat, S', G2M')

# cell-level agreement
SamePhase = mean_i [ y[i] == y_hat[i] ]
```

## A.4 EXPERIMENTS

Table 5: End-to-end reconstruction hyperparameter search space. We sweep MLP depth, width, latent dimension, and (for VAE models) the KL weight $\beta$. Best model is selected by reconstruction loss (i.e., MSE) on validation set and metrics are reported on test set.

| Component | Values explored |
|---|---|
| Model families | PCA, AE, VAE |
| Encoder/decoder layers $L$ | $\{1, 2, 3, 4\}$ |
| Hidden units per layer $H$ | $\{1024, 2048, 4096\}$ |
| Latent dimension $d$ | $\{10, 32, 128, 512, 2048\}$ |
| KL weight (VAE only) $\beta$ | $\{0.1, 1.0\}$ |

Table 6: Decoder-only reconstruction hyperparameter search space. The encoder is frozen (foundation model), and we sweep decoder depth and width. Best model is selected by reconstruction loss (i.e., MSE) on validation set and metrics are reported on test set.

| Component | Values explored |
|---|---|
| Frozen encoders | scGPT, SCimilarity, STATE, scConcept |
| Decoder families | MLP decoder |
| Decoder layers $L$ | $\{1, 2, 3, 4\}$ |
| Hidden units per layer $H$ | $\{1024, 2048, 4096\}$ |

## A.5 ADDITIONAL RESULTS

### A.5.1 END-TO-END RECONSTRUCTION PERFORMANCE

Table 7: End-to-end reconstruction performance summary of $d = 10$ (mean $\pm$ 1 std across dataset$\times$split settings; $n = 9$ for Overall).

| Model | Overall | Statistical | | | Biological | | | |
|---|---|---|---|---|---|---|---|---|
| | | MSE | $R^2$ | E-distance | Pathway Sp. | Coexpr. Sp. | Cell-cycle | DEG recall |
| AE-N | **0.83** ± 0.06 | **0.11** ± 0.04 | 0.90 ± 0.07 | **2.19** ± 2.02 | **0.52** ± 0.09 | **0.86** ± 0.02 | 0.54 ± 0.19 | **0.35** ± 0.24 |
| AE-M | 0.78 ± 0.07 | 0.11 ± 0.04 | 0.90 ± 0.07 | 2.30 ± 2.15 | 0.52 ± 0.09 | 0.84 ± 0.02 | **0.55** ± 0.20 | 0.31 ± 0.26 |
| AE-O | 0.81 ± 0.05 | 0.10 ± 0.04 | **0.90** ± 0.07 | 2.22 ± 2.16 | 0.52 ± 0.10 | 0.83 ± 0.02 | 0.55 ± 0.20 | 0.32 ± 0.26 |
| scVI-N | 0.50 ± 0.08 | 0.12 ± 0.04 | 0.83 ± 0.12 | 4.52 ± 3.36 | 0.36 ± 0.07 | 0.84 ± 0.02 | 0.52 ± 0.15 | 0.18 ± 0.13 |
| scVI-M | 0.35 ± 0.04 | 0.12 ± 0.05 | 0.81 ± 0.13 | 5.35 ± 4.10 | 0.36 ± 0.07 | 0.82 ± 0.04 | 0.52 ± 0.14 | 0.16 ± 0.14 |
| scVI-O | 0.50 ± 0.08 | 0.11 ± 0.04 | 0.84 ± 0.11 | 4.25 ± 3.14 | 0.36 ± 0.06 | 0.81 ± 0.02 | 0.52 ± 0.14 | 0.16 ± 0.14 |
| PCA | 0.24 ± 0.06 | 0.13 ± 0.05 | 0.67 ± 0.07 | 9.23 ± 4.11 | 0.39 ± 0.07 | 0.78 ± 0.04 | 0.51 ± 0.18 | 0.25 ± 0.28 |

Table 8: End-to-end reconstruction performance summary of $d = 32$ (mean $\pm$ 1 std across dataset$\times$split settings; $n = 9$ for Overall).

| Model | Overall | Statistical | | | Biological | | | |
|---|---|---|---|---|---|---|---|---|
| | | MSE | $R^2$ | E-distance | Pathway Sp. | Coexpr. Sp. | Cell-cycle | DEG recall |
| AE-N | **0.82 ± 0.08** | 0.09 ± 0.04 | 0.92 ± 0.06 | **1.22 ± 1.23** | **0.65 ± 0.11** | **0.87 ± 0.01** | 0.57 ± 0.19 | **0.34 ± 0.26** |
| AE-M | 0.79 ± 0.09 | 0.10 ± 0.04 | 0.91 ± 0.06 | 1.54 ± 1.52 | 0.63 ± 0.10 | 0.86 ± 0.02 | **0.57 ± 0.19** | 0.31 ± 0.27 |
| AE-O | 0.75 ± 0.05 | **0.09 ± 0.04** | **0.92 ± 0.06** | 1.31 ± 1.34 | 0.64 ± 0.11 | 0.86 ± 0.02 | 0.57 ± 0.18 | 0.31 ± 0.27 |
| scVI-N | 0.47 ± 0.08 | 0.11 ± 0.04 | 0.84 ± 0.11 | 3.71 ± 2.89 | 0.43 ± 0.06 | 0.85 ± 0.02 | 0.55 ± 0.17 | 0.19 ± 0.15 |
| scVI-M | 0.35 ± 0.09 | 0.12 ± 0.05 | 0.81 ± 0.16 | 5.71 ± 5.92 | 0.43 ± 0.06 | 0.84 ± 0.03 | 0.54 ± 0.16 | 0.17 ± 0.13 |
| scVI-O | 0.45 ± 0.08 | 0.11 ± 0.05 | 0.85 ± 0.10 | 3.62 ± 2.69 | 0.43 ± 0.05 | 0.83 ± 0.02 | 0.54 ± 0.16 | 0.18 ± 0.14 |
| PCA | 0.36 ± 0.08 | 0.12 ± 0.04 | 0.77 ± 0.10 | 4.34 ± 3.64 | 0.54 ± 0.07 | 0.84 ± 0.04 | 0.54 ± 0.17 | 0.28 ± 0.28 |

Table 9: End-to-end reconstruction performance summary of $d = 512$ (mean $\pm$ 1 std across dataset$\times$split settings; $n = 9$ for Overall).

| Model | Overall | Statistical | | | Biological | | | |
|---|---|---|---|---|---|---|---|---|
| | | MSE | $R^2$ | E-distance | Pathway Sp. | Coexpr. Sp. | Cell-cycle | DEG recall |
| AE-N | 0.73 ± 0.13 | 0.05 ± 0.03 | **0.95 ± 0.05** | 0.55 ± 0.58 | 0.90 ± 0.08 | 0.94 ± 0.02 | 0.65 ± 0.18 | 0.29 ± 0.29 |
| AE-M | **0.81 ± 0.06** | **0.05 ± 0.03** | 0.95 ± 0.04 | 0.50 ± 0.47 | **0.90 ± 0.07** | **0.94 ± 0.02** | **0.67 ± 0.19** | 0.29 ± 0.29 |
| AE-O | **0.81 ± 0.06** | 0.05 ± 0.04 | 0.95 ± 0.05 | **0.49 ± 0.47** | 0.89 ± 0.08 | 0.93 ± 0.02 | 0.67 ± 0.19 | **0.32 ± 0.28** |
| scVI-N | 0.43 ± 0.05 | 0.09 ± 0.04 | 0.88 ± 0.08 | 2.12 ± 2.00 | 0.67 ± 0.08 | 0.91 ± 0.03 | 0.61 ± 0.18 | 0.20 ± 0.12 |
| scVI-M | 0.25 ± 0.08 | 0.10 ± 0.05 | 0.85 ± 0.09 | 3.86 ± 3.20 | 0.59 ± 0.11 | 0.89 ± 0.06 | 0.58 ± 0.18 | 0.19 ± 0.13 |
| scVI-O | 0.39 ± 0.06 | 0.09 ± 0.04 | 0.88 ± 0.06 | 2.47 ± 1.75 | 0.65 ± 0.07 | 0.90 ± 0.03 | 0.61 ± 0.17 | 0.20 ± 0.13 |
| PCA | 0.59 ± 0.06 | 0.07 ± 0.04 | 0.90 ± 0.07 | 0.68 ± 1.30 | 0.84 ± 0.07 | 0.92 ± 0.03 | 0.62 ± 0.21 | 0.29 ± 0.28 |

Table 10: End-to-end reconstruction performance summary of $d = 2048$ (mean $\pm$ 1 std across dataset$\times$split settings; $n = 9$ for Overall).

| Model | Overall | Statistical | | | Biological | | | |
|---|---|---|---|---|---|---|---|---|
| | | MSE | $R^2$ | E-distance | Pathway Sp. | Coexpr. Sp. | Cell-cycle | DEG recall |
| AE-N | 0.77 ± 0.11 | **0.02 ± 0.02** | **0.97 ± 0.03** | 0.31 ± 0.32 | **0.97 ± 0.03** | **0.96 ± 0.00** | 0.74 ± 0.12 | 0.34 ± 0.28 |
| AE-M | 0.57 ± 0.13 | 0.03 ± 0.02 | 0.95 ± 0.04 | 0.45 ± 0.49 | 0.95 ± 0.04 | 0.91 ± 0.03 | 0.71 ± 0.15 | 0.34 ± 0.25 |
| AE-O | 0.72 ± 0.13 | 0.03 ± 0.03 | 0.96 ± 0.05 | 0.33 ± 0.35 | 0.96 ± 0.04 | 0.92 ± 0.02 | 0.76 ± 0.12 | **0.36 ± 0.24** |
| scVI-N | 0.46 ± 0.04 | 0.06 ± 0.03 | 0.93 ± 0.06 | **1.20 ± 1.15** | 0.84 ± 0.06 | **0.95 ± 0.01** | **0.69 ± 0.13** | 0.20 ± 0.12 |
| scVI-M | 0.26 ± 0.12 | 0.08 ± 0.05 | 0.88 ± 0.07 | 2.97 ± 2.47 | 0.73 ± 0.14 | 0.91 ± 0.04 | 0.65 ± 0.17 | 0.18 ± 0.13 |
| scVI-O | 0.36 ± 0.06 | 0.07 ± 0.04 | 0.90 ± 0.06 | 1.92 ± 1.56 | 0.79 ± 0.09 | 0.92 ± 0.02 | 0.68 ± 0.13 | 0.19 ± 0.13 |
| PCA | **0.86 ± 0.09** | 0.02 ± 0.02 | 0.97 ± 0.04 | **0.15 ± 0.32** | 0.97 ± 0.03 | 0.96 ± 0.01 | **0.77 ± 0.15** | 0.33 ± 0.28 |

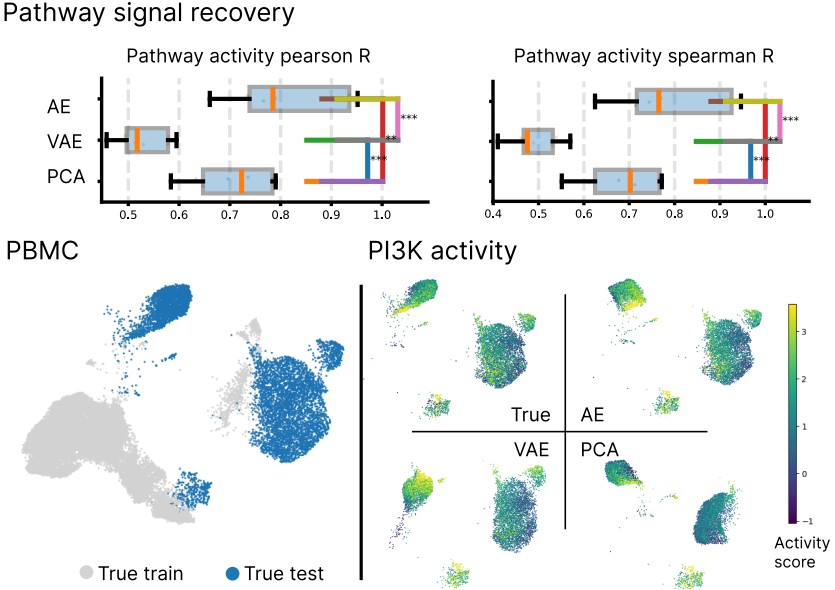

Figure 4: Pathway activity recovery from end-to-end reconstruction on *PBMC-10M* under cell-type OOD.

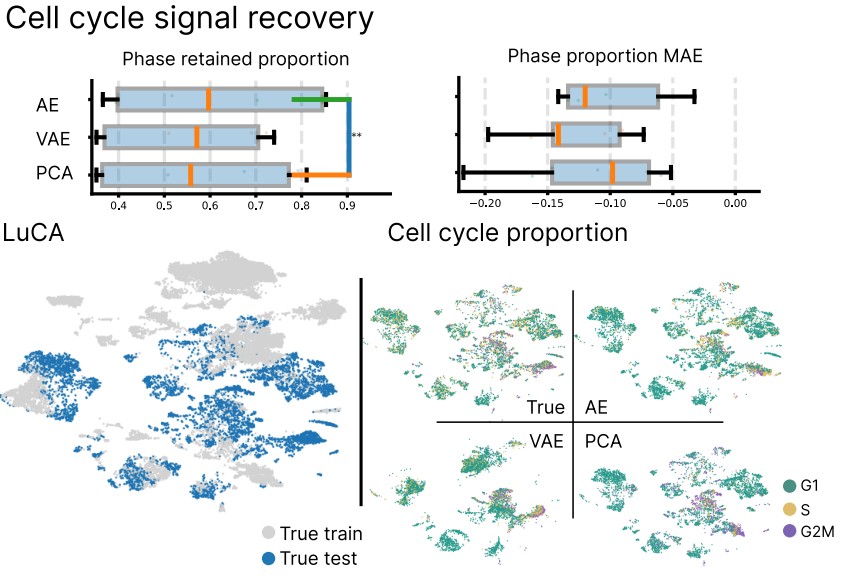

Figure 5: Cell cycle signal recovery from end-to-end reconstruction on *LuCA*

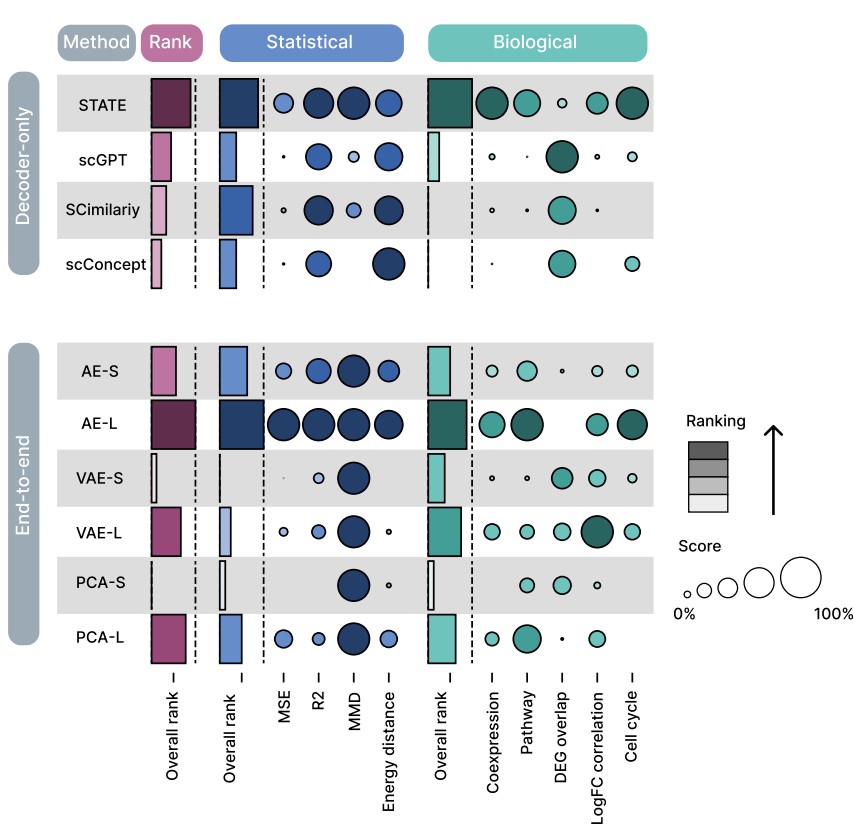

Figure 6: End-end and Decoder-only comparison on *PBMC-10M* - Split 1 ( the most OOD level split)

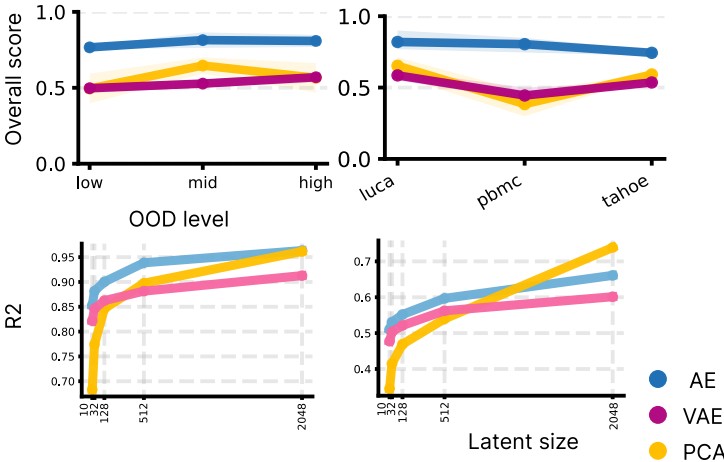

Figure 7: **Robustness to OOD and latent scaling. Top:** Overall performance across OOD levels (left) and across datasets (right), highlighting robustness to distribution shift. **Bottom:** Effect of latent size on performance, shown for a statistical metric ($R^2$, left) and a biological metric (DEG recovery, right).

