# OpenReview forum: "What information is preserved in latent cell embeddings? A Benchmark for Single-Cell Reconstruction"
_ICLR.cc/2026/Workshop/LMRL — ICLR 2026 Workshop LMRL Poster_

### Official Review · Reviewer_pu2z · 2026-02-25
**A Practical Reconstruction Benchmark for Single-Cell Foundation-Model Embeddings**

**Rating:** 8
**Confidence:** 3

**Review:**

## Summary

This paper argues that reconstruction ability is a necessary (and often under-validated) property of latent representations used for latent-shift modeling in single-cell genomics, especially for “virtual cell” workflows where biological interpretation ultimately happens in gene-expression space.

The authors propose a systematic reconstruction benchmark covering both (i) end-to-end learned representations (PCA, AE, VAE/scVI) and (ii) decoder-only reconstruction from frozen foundation-model embeddings (scGPT, SCimilarity, STATE, scConcept) using a trained decoder head.

They evaluate across three large and diverse datasets (Tahoe-100M, PBMC-10M, LuCA) spanning perturbational and observational regimes.

Performance is assessed using a mix of statistical fidelity (e.g., MSE/R²/MMD) and downstream-inspired biological metrics (DEG recovery, logFC correlation, pathway activity, coexpression, cell-cycle preservation), plus an aggregate rank score.

## Strengths

Treating representation choice as a first-order modeling decision (rather than an implementation detail) is very compelling and inspiring to the field.

Metrics are biologically grounded, which go beyond MSE and evaluate whether reconstructions preserve signals relevant to common analyses such as differential expression and pathway programs.

The multi-split design (varying OOD difficulty across datasets) strengthens the conclusions and makes the benchmark more practically useful.

I think this paper is well-motivated, the benchmark is thoughtfully designed, and the results provide practical, actionable guidance

## Limitations and questions

Using a consistent MLP decoder is reasonable, but the conclusion “how much can be decoded from this embedding” inevitably depends on decoder expressivity and inductive bias. A brief exploration on decoder families (e.g., shallow vs deeper, linear vs nonlinear) would help interpret whether gaps are embedding-limited or decoder-limited.

---

### Official Review · Reviewer_ztDN · 2026-02-26

**Rating:** 5
**Confidence:** 5

**Review:**

This paper presents a benchmark for evaluating latent representations for single-cell transcriptomic data through their ability to support reconstruction and perturbation aware recovery of gene expression profiles. I think the problem is interesting and relevant, especially for latent-shift and virtual-cell settings where decoding back into gene space is important. The benchmark is also thoughtfully constructed in several respects, including the use of multiple datasets, multiple OOD regimes, and biological evaluation metrics beyond pure reconstruction error, such as differential expression, pathway activity, coexpression, and cell-cycle preservation.

My main concern is with the interpretation of the foundation-model results. In the decoder-only setting, the paper evaluates pretrained foundation models as frozen encoders and trains only a decoder on top, explicitly to reflect common transfer usage. This is a reasonable protocol for measuring how much gene-level information is decodable from frozen embeddings, but I do not think it supports broad conclusions about the overall quality of those latent spaces or whether they “encode meaningful biology.” These models were generally not trained for reconstruction, and the paper itself acknowledges that evaluating fine-tuned foundation models is an important next step.  As a result, the current benchmark seems better suited to answer a narrower question, how suitable frozen embeddings are for reconstruction-centric latent-shift workflows, than a broader question about the intrinsic biological quality of foundation-model representations.

Relatedly, I am not fully convinced by the paper’s central assumption that stronger reconstruction necessarily implies a more biologically meaningful latent space. Reconstruction is certainly relevant for applications that must decode back into expression space, and the authors motivate this well in the context of latent-shift modeling. However, biological usefulness can also include discriminative, functional, or transfer properties that are not well captured by reconstruction-based evaluation alone. For that reason, I think the paper should either soften its claims about “meaningful biology” or complement the benchmark with tasks that probe biological alignment more directly from the embeddings.

A secondary concern is the OOD definition. The benchmark uses label-based held-out units such as cell type, donor, study, and condition to define increasingly difficult OOD splits.  This is reasonable and transparent, but it may not fully capture transcriptomic novelty. In single-cell settings, OOD difficulty often depends on similarity in the underlying expression space rather than metadata labels alone. The paper would be stronger if it also quantified train-test separation using transcriptomic distance or a related similarity-based measure, to better characterize how challenging each split actually is.

Overall, I think the paper addresses an interesting question and provides a potentially useful benchmark, but I am not yet convinced that the current experimental design justifies the broader claims being made about foundation-model latent spaces. I would be more positive if the claims were narrowed to frozen-embedding reconstructability, and if the authors either included fine-tuned FM baselines or more clearly separated reconstruction suitability from broader biological representation quality.

---

### Meta-Review · Area_Chair_xSyT · 2026-02-25

**Recommendation:** Accept (Poster)
**Confidence:** 4

**Metareview:**

Accept

---

### Decision · Program_Chairs · 2026-03-02

**Decision:**

Accept (Poster)

**Comment:**

Please see the meta-review.